# Evolutionary pathways in soil-landscape evolution models

W. Marijn van der Meij[1]

[1]Institute of Geography, University of Cologne, Albertus-Magnus-Platz, 50923 Cologne, Germany

*Correspondence to*: W. Marijn van der Meij (m.vandermeij@uni-koeln.de)

**Abstract**

Soils and landscapes can show complex, non-linear evolution, especially under changing climate or land use. Soil-landscape evolution models (SLEMs) are increasingly equipped to simulate the development of soils and landscapes over long timescales under these changing drivers, but provide large data output that can be difficult to interpret and communicate. New tools are required to analyze and visualize large model outputs.

In this work, I show how spatial and temporal trends in previously published model results can be analyzed and visualized with evolutionary pathways, which are possible trajectories of the development of soils. Simulated differences in rainfall and land use control progressive or regressive soil development and convergence or divergence of the soil pattern. These changes are illustrated with real-world examples of soil development and soil complexity.

The use of evolutionary pathways for analyzing the results of SLEMs is not limited to the examples in this paper, but they can 15 be used on a wide variety of soil properties, soil pattern statistics and models. With that, evolutionary pathways provide a promising tool to analyze and visualize soil model output, not only for studying past changes in soils, but also for evaluating future spatial and temporal effects of soil management practices in the context of sustainability.

**Keywords**: soil-landscape evolution model, evolutionary pathways, soil development, sustainable soil management, 20 complexity

# 1    Introduction

Soils are natural resources that provide valuable functions such as food provision and carbon storage (Adhikari and Hartemink, 2016). Due to intensive land management, these resources are threatened and the resources decline. Understanding how soil and landscape properties are affected by anthropogenic pressure is essential not only to assess the impact of intensive land
management on soil functions, but also to develop sustainable management strategies where natural processes can be used to improve the soil functions and ecosystem functions, such as carbon sequestration (Dominati et al., 2010; Minasny et al., 2017). The development, and degradation, of soils is not only something of recent times. Starting with first agriculture and increasing in intensity towards the present, humans have triggered and increased erosion and degradational processes (Dotterweich, 2013; Stephens et al., 2019). A longer-term perspective of centuries to millennia is thus required to understand and describe the
dynamics of soil systems in response to anthropogenic pressure, as these time scales capture the changes in inherent soil properties such as soil texture and terrain position, that form the basis for more dynamic manageable soil properties, such as carbon and nutrient stocks (Dominati et al., 2010).

Through time, soil development can follow multiple trajectories, depending on the initial state and the driving forces. A change in land use, for example, will affect soil development differently for different topographic positions (Sommer et al., 2008).
These developmental trajectories of soil-landscape systems can be described in a conceptual and quantitative way using evolutionary pathways, which were developed and extensively described by professor Jonathan D. Phillips (e.g., Phillips, 2019). A major problem in quantifying these pathways is the need for a large set of spatial and especially temporal soil data. This is often problematic, because measurements only cover a brief period of time. In areas where a longer chronological record is present, such as chronosequences, the number of data points is often limited and it is impossible to sample the exact
location multiple times without disturbing soil development.

Soil-landscape evolution models (SLEMs) can provide a solution here. SLEMs are numerical models that simulate a set of pedogenic and geomorphic processes that affect the spatial and temporal development of soils and landscapes (Minasny et al., 2015). Recent developments in SLEMs enable a more accurate simulation of driving forces behind soil and landscape development (e.g., Van der Meij et al., 2018) and with that they become increasingly equipped to simulate soil and landscape
development in a wide variety of settings, over long timescales (Bouchoms et al., 2017; Bock et al., 2018; Welivitiya et al., 2019; Van der Meij et al., 2020; Kwang et al., 2022). These developments make SLEMs suitable not only to study past soil-landscape dynamics and changes in inherent properties, but also to evaluate the effect of future changes in climate and land use on manageable soil and landscape properties, such as carbon and nutrient stocks (Minasny et al., 2015). SLEMs can provide a large amount of spatiotemporal data on soil and landscape properties to calculate the evolutionary pathways. Conversely,
evolutionary pathways might be a suitable tool to analyze and visualize the large data output from SLEMs and characterize the development of soil and landscape properties under different drivers.

In this study, I test the concept of evolutionary pathways as a new tool to analyze and visualize trends in the results of SLEMs.

## 2 Methods

### 2.1 Model study

This study builds further on the results of Van der Meij et al. (2020), who used their SLEM called HydroLorica to simulate the development of soils and landscapes under different rainfall and land use scenarios. A detailed description of the methodology and results is provided in their paper and supplementary information. Here I will briefly repeat the key aspects of the study, that are of importance to this study as well.

HydroLorica is a reduced-complexity soil-landscape evolution model. The surface of the landscape is represented by a raster-
based digital elevation model (DEM). Below each raster cell, there are 25 vertically stacked layers, representing the soils (Figure 1). Inside these layers, the model keeps track of five texture classes and two organic matter classes. HydroLorica simulates a set of geomorphic, pedogenic, hydrologic and biotic processes that mix or transport contents of the soil layers (Figure 1). Changes in the composition of the layers lead to changes in layer thickness that propagate to changes in surface elevation. The main difference between HydroLorica and other SLEMs is that it simulates the surface water balance spatially
explicit (Van der Meij et al., 2018). Spatial differences in overland flow and infiltration directly control the rates of water erosion and clay translocation. The water stress, determined with a modified Budyko Curve, is used to determine vegetation type (grass or forest). The vegetation type in turn controls rates of bioturbation, soil creep and soil organic matter cycling, and determines water erosion protection by vegetation and the occurrence of tree throw. These processes are thus indirectly controlled by water flow and availability.

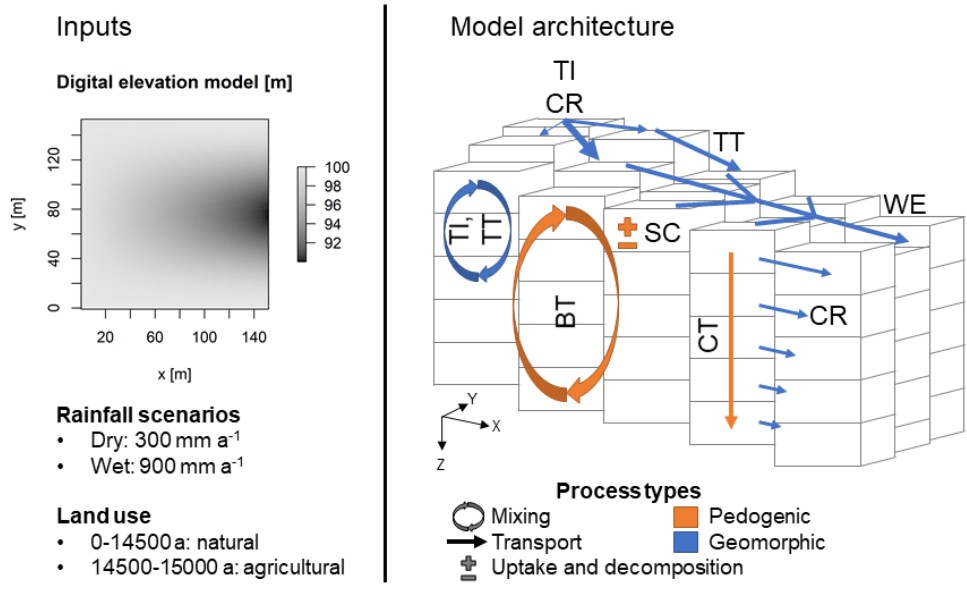


Figure 1: Overview of inputs (left) and architecture and processes (right) for the modelling study. The abbreviated process names are CR: creep, WE: water erosion and sedimentation, TT: tree throw, TI: tillage, BT: bioturbation, CT: clay translocation, SC: soil organic matter cycling. The colors of the symbols indicate if it is a geomorphic or pedogenic process.

Water erosion is simulated as a grain-size dependent advective process. Uptake and deposition of sediments are determined by the transport capacity, which is a function of local slope and volume of surface water flow (stream power equation, Temme and Vanwalleghem, 2016). Soil creep and tillage are slope-dependent diffusive processes, that gradually smoothen the topography. Transport of soil material by creep is occurring in all soil layers, with an exponentially decreasing rate with soil depth. Transport by tillage occurs at the surface of the soils. Next to transport, tillage also homogenizes the soil layers that are

in the reach of the ploughing depth. Tree throw is simulated as a random event-based process that transports soil material inside the root clump over a certain transport distance, that depends on the slope of the terrain. Before deposition, the soil material is homogenized. Clay translocation is simulated as an advection-diffusion equation (Jagercikova et al., 2015), where the advective process transports clay downward in the soil profile. The rate of clay translocation is controlled by the infiltration rate. Clay translocation can occur everywhere in the landscape. The only limitation on clay translocation is the amount of clay

that is available for transport. This water-dispersible fraction was estimated using equations from Brubaker et al., (1992). The diffusive process of clay translocation is represented by bioturbation, that slowly mixes soil layers (Van der Meij et al., 2020). Soil organic matter cycling is simulated using an uptake and a decomposition function. The uptake of SOM depends on vegetation type (grassland, forest or cereals). There is a division between quickly and slowly decomposing organic matter, that is controlled with a humification fraction (Yoo et al., 2006; Temme and Vanwalleghem, 2016). All soil process rates decrease

with soil depth, following an exponential decay function. Weathering processes were not included in this simplified representation of the soil-landscape system.

The original study simulated three rainfall scenarios: dry (P = 300 mm a$^{-1}$), humid (P = 600 mm a$^{-1}$) and wet (P = 900 mm a$^{-1}$). In this study, I will focus on the dry and wet scenario, where the main natural vegetation types are grasslands and forests, respectively. The simulations were performed in a small artificial catchment as spatial setting to avoid effects of local

idiosyncrasies (Figure 1). The parent material is homogeneous loess material of 15% sand, 75% silt and 10% clay. The model runs were 15000 years with an annual timestep. In the first 14500 years of the simulations there was natural soil and landscape development. In the last 500 years of the simulations, anthropogenic effects were introduced in the form of tillage erosion and change of vegetation type to cereals. These periods loosely reflect Holocene soil development and intensive land management starting after the Middle Ages.

In the results I will focus on the following soil properties: soil organic matter (SOM) stocks [kg m$^{-2}$], which represents a relatively fast changing, manageable property, and the depth to Bt horizons [m], which represent a relatively slow changing, inherent property (Dominati et al., 2010). SOM stocks are defined as the total sum of SOM in a vertical soil profile. Depth to Bt horizons is defined as the depth at which the clay fraction first exceeds the clay fraction of the parent material.

## 2.2 Evolutionary pathways

I calculated two types of evolutionary pathways to quantify the development of the soil properties: soil complexity and soil-landscape development stage (Figure 2). Soil development is spatially variable, because its drivers are spatially variable as well. Therefore, I used the spatial average of a soil property x to measure the stage of soil property development in the entire

landscape. I named this average development the soil-landscape development stage (SLDS). Changes in the SLDS can follow progressive and regressive pathways (Johnson and Watson-Stegner, 1987; Phillips, 1993; Sommer et al., 2008; Sauer, 2015).

Progressive development indicates forward development of a property, such as soil deepening and carbon uptake, while regressive development indicates reduction of a certain property, such as soil loss by erosion and carbon loss. I calculated the SLDS by taking the average level of development of a certain soil property $x$ in space at a certain moment $t$ in time (Eq. (1)). Positive changes in SLDS over time indicate progressive evolution, while negative changes indicate regressive evolution (Figure 2).

$$\Delta SLDS_{x,t} = \frac{mean(x)_t - mean(x)_{t-\Delta t}}{\Delta t} \tag{1}$$

Soil complexity describes the level of variation or heterogeneity of soil and terrain properties in space (Phillips, 2017). Changes in soil complexity are driven by changes in soil forming factors, nonlinearity in pedogenic processes, internal feedbacks in the soil-landscape system and anthropogenic disturbances. I calculated the complexity as the standard deviation of a certain soil property $x$ in space at a certain moment $t$ in time (Eq. (2)). Positive changes in complexity over time indicate divergent

evolution, while negative changes indicate convergent evolution of the soil property (Figure 1, Temme et al., 2015; Phillips, 2017).

$$\Delta complexity_{x,t} = \frac{sd(x)_t - sd(x)_{t-\Delta t}}{\Delta t} \tag{2}$$

The evolutionary pathways were calculated for timespans $\Delta t$ of 50 years. This relatively small timespan was selected to provide more detail in the initial phase of natural soil development and in the agricultural phase. The units of SLDS and soil complexity

are the same as the soil property they are calculated for. The units of these quantities changing over time, resulting from Eqs (1) and (2), are kg m$^{-2}$ a$^{-1}$ for SOM stocks and m a$^{-1}$ for depth to the Bt horizons.

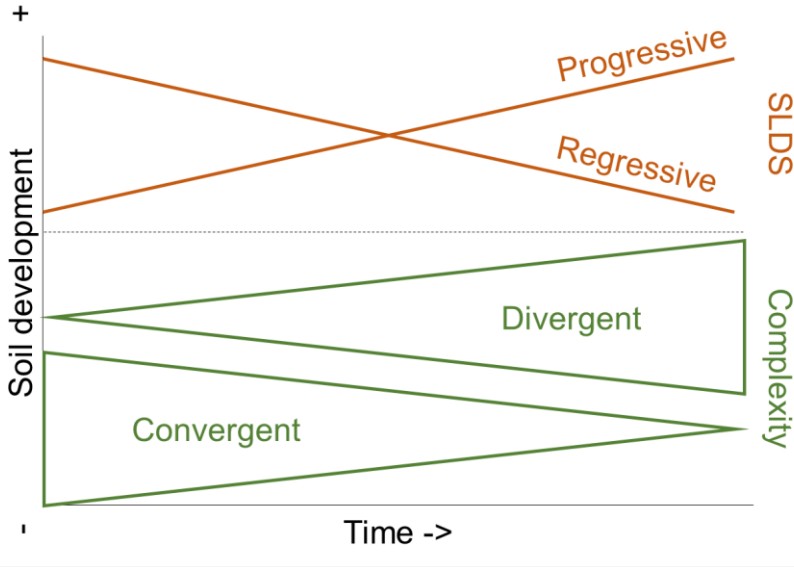

**Figure 2: Illustration of evolutionary pathways related to soil-landscape development state (SLDS) and soil complexity.**

## 3    Results

Figure 3 shows the development of SLDS and soil complexity, i.e. the evolutionary pathways, of the studied soil properties over time. The depth to Bt horizons shows large differences between the dry and wet scenarios in the natural phase (Figure 3A), with both higher average values (higher SLDS) and larger standard deviation (higher complexity) for the wet scenario. After the start of the agricultural phase, for the dry scenario, the pattern of depth to Bt shows continuous increasing complexity and, after an initial drop in SLDS, slightly fluctuating regressive and progressive pedogenesis. For the wet scenario, the complexity initially decreases and starts to increase again 100 years into the agricultural phase, while the SLDS is continuously decreasing.

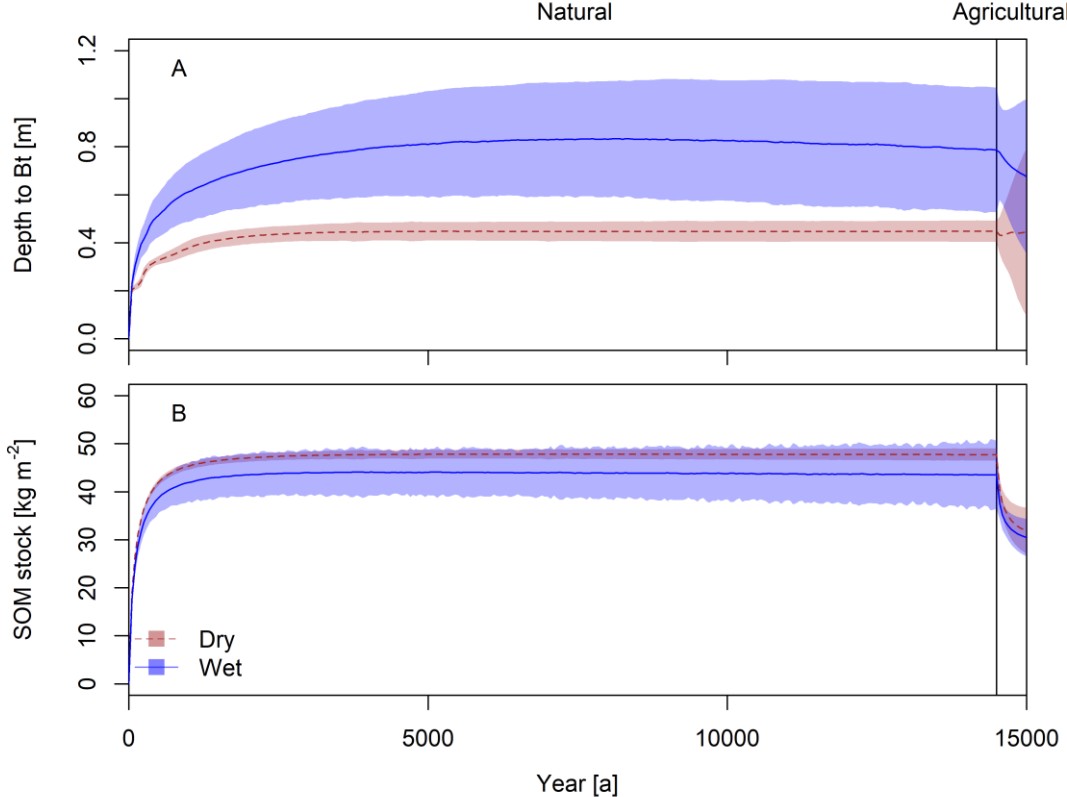

**Figure 3: Evolution of the depth to Bt horizons (A) and the SOM stocks (B) in the simulations with HydroLorica. The line indicates the average (SLDS) and the polygon indicates the standard deviation (complexity) of the property. The vertical line at year 14500 indicates the start of the agricultural phase.**

The SOM stocks show smaller differences between the two scenarios, compared to the depth to Bt horizons (Figure 3B). In the natural phase, the dry scenario shows SOM stocks with higher SLDS and lower complexity than the wet scenario. In the wet scenario, the complexity shows a slight overall increase over time, indicating mainly divergent evolution. Over shorter timespans, the complexity fluctuates slightly. In the agricultural phase, SOM stocks in both scenarios drop to a lower level.

Figure 4 further specifies the evolutionary pathways in the model results. The pathways are displayed in quadrants, where each quadrant is a unique combination of changes in complexity and SLDS. The points in each quadrant show changes in complexity and SLDS over a time span of 50 years. The lines that connect the points represent the evolutionary pathways via which the soil properties evolve. When the points converge to the origin of the graph, the soil pattern reaches a steady state in its development. When the points converge to a point elsewhere in the graph, the soil pattern has a steady rate of change.

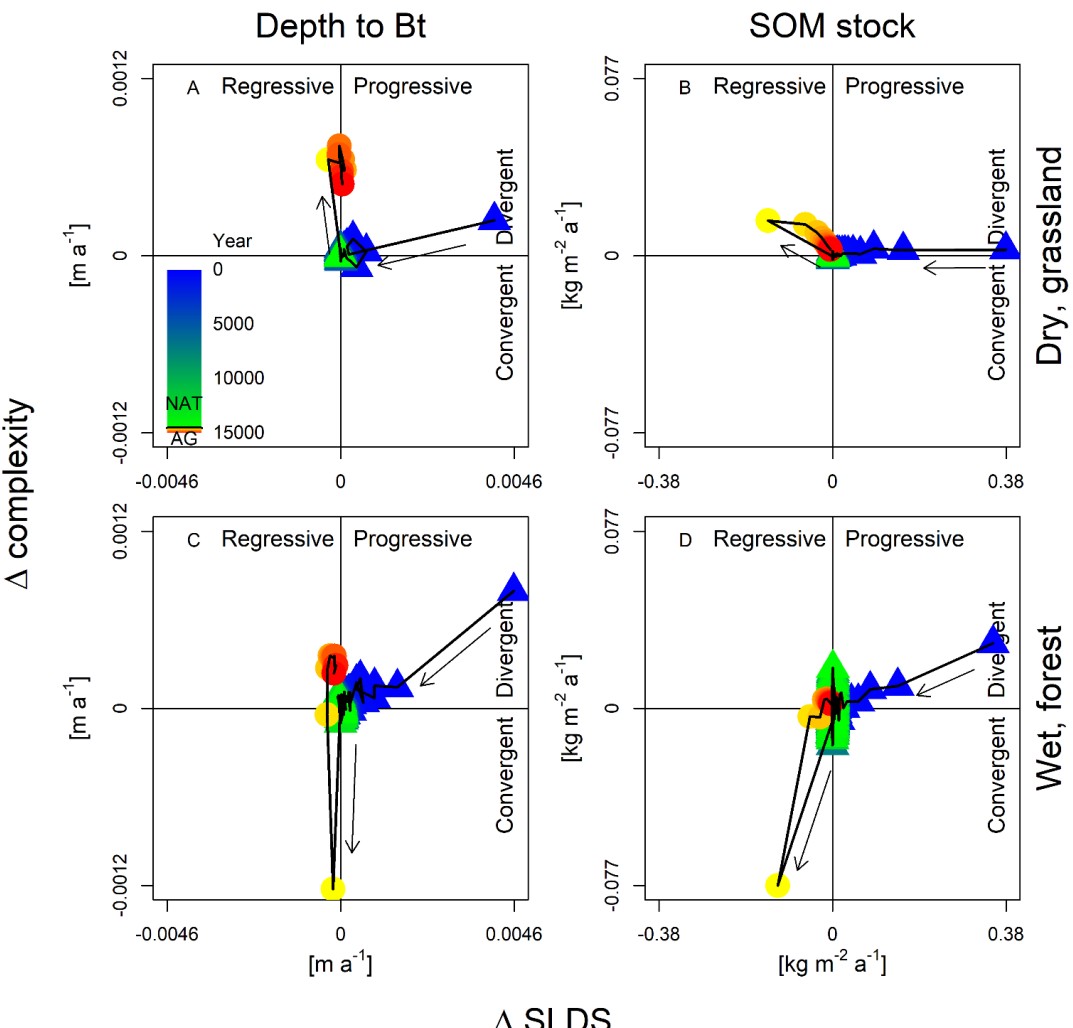


**Figure 4: Changes in SLDS (X-axis) and complexity (Y-axis) for depth to Bt horizons (A, C) and SOM stocks (B, D) for both rainfall scenarios (dry: A, B; wet: C, D). The colors of the symbols indicate the year. The symbols themselves indicate the land use (triangles: natural, year 0-14500, blue to green colors; circles: agricultural, year 14500-15000, yellow to red colors). The timestep ▲t over which the evolutionary pathways are calculated is 50 years for both land uses.**

Natural evolutionary pathways of both soil properties in the dry scenario converge to a steady state at the origin of the graphs (Figure 4A&B). Both pathways are dominantly progressive and there is little change in the complexity of the soil patterns. The

natural evolutionary pathways in the wet scenario show much more divergent development for both properties, compared to the dry scenario (Figure 4C&D). The depth to Bt horizons in the wet scenario converges slower towards a steady state, compared to the dry scenario. Both properties in the wet scenario do not reach a steady state. Instead, the points remain scattered around the origin of the graph, with fluctuating convergent and divergent development. This can be seen by the larger spread in green-colored triangles, that indicate later stages in the natural development. The SLDS remains constant during these fluctuations. This is especially visible for the SOM stocks (Figure 4D).

In the natural phase, the largest changes in both soil properties occur at the start of soil development (Figure 4). This is mainly an increase in the SLDS, or progressive development of the properties. In the agricultural phase, the first step after cultivation shows large changes in the complexity of the soil properties. For all scenarios and properties, the change in complexity after cultivation exceeds any natural changes in complexity. Depending on the climatic setting, the initial change in complexity is either divergent (dry) or convergent (wet). The pathways for depth to Bt horizons stay or turn divergent after the initial change. For the SOM stocks, the pathways approach a stable complexity. After cultivation, all properties and scenarios initially show regressive soil development, which indicates that overall, soils have less SOM and shallower Bt horizons. However, the changes are minor compared to changes at the start of the natural phase. Changes in SLDS after cultivation for the depth to Bt horizons are almost zero. All properties approach a stable SLDS during the agricultural phase. Depending on the changes in complexity, the properties reach a new steady state (SOM stocks) or have a relatively steady rate of change (depth to Bt).

# 4 Discussion

## 4.1 Evolutionary pathways

The model results show the effects of rainfall and land use on soil pattern development. There are clear differences between the two rainfall scenarios, which can be easily distinguished with the evolutionary pathways. Higher rainfall has three main effects on the soils and landscape in the model: 1) it causes more overland flow and higher water erosion, 2) it leads to higher infiltration rates and consequently more intense clay translocation, and 3) it facilitates tree growth and consequently tree throw, which is a process that constantly changes the local spatial variation of the soil properties (Phillips, 2001; Šamonil et al., 2018).

These processes all operate on different spatial scales and can have opposite effects, which increases the complexity of the soil properties and leads to more divergent soil development, compared to the dry scenario (Figure 3, Figure 4). The rates of local perturbations, such as tree throw, can exceed the capacity of pedogenic processes to respond to these perturbations (Phillips, 2017). This may prevent the reaching of a steady state, where the SLDS and complexity remain constant. This is for example visible for both soil properties in the wet scenario (Figure 3, Figure 4), where the evolutionary pathways take longer to approach

a steady state, while continuously changing the complexity of the pattern. In the end, the steady state is not reached, because the disturbance cycle of tree throw is faster than the time the soil properties need to recover from the disturbance. The effects of these disturbances seem larger for the SOM stocks than for the depth to Bt horizons. This can be, because the highest values of SOM stocks occur in the topsoil, which is also the part of the soil that is most actively disturbed by different processes. For the dry scenario, in the absence of tree throw, a steady state was reached for both properties. The model simulations show that

a steady state in soil development can indeed be reached, but only under stable circumstances with little to no external perturbations, which supports statements that a steady states in soils are a rare occurrence (Phillips, 2010; Sauer, 2015).

The introduction of tillage and agricultural crops had a huge impact on the development of the soil patterns in the model. Overall, SOM stocks went down due to a lower input and increased erosion, leading to regressive development of the SOM stocks (Figure 3B). The average depth to Bt horizons also followed a slightly regressive pathway (Figure 3A). Whether the

initial response of both soil properties to anthropogenic forcing was divergent or convergent, differed between the rainfall scenarios. If the natural soil pattern was already complex, such as the in wet scenario, initial tillage erosion and land management reduced this complexity (Figure 3). In the dry scenario, where the natural complexity was much lower, the tillage erosion caused divergent evolution of the soil properties. The response of evolutionary pathways to anthropogenic forcing thus depends on the properties of the natural soil pattern. After this initial change, all evolutionary pathways stayed or turned

divergent and approached a new steady state for SOM (Figure 4B&D) or approached a steady rate of change for the depth to Bt (Figure 4A&C).

The largest changes in the development of the soil properties occurred in the first 50 years of the natural phase (Figure 4). These findings conform to those from other studies, where initial soil development occurs with high rates, because the soil system is far from equilibrium and initial soils contain more reactive surfaces than soils in older settings (Dümig et al., 2011;

Kabala and Zapart, 2012; Elmer et al., 2013). The largest changes in complexity occur in the first 50 years of the agricultural

phase, where the changes in complexity exceed those of any natural soil development. Changes in complexity remain relatively large for the depth to Bt horizons, while changes in complexity decrease over time for the SOM stocks. These findings support that humans are the dominant soil forming factor in agricultural landscapes (Amundson and Jenny, 1991; Dudal, 2005; Richter and Yaalon, 2012), not only by elevating erosion rates above natural levels (Wilkinson, 2005; Nearing et al., 2017), but also

by severely modifying the configuration and complexity of soil patterns.

It is important to note here that the selected timespan over which the evolutionary pathways were calculated has a large effect on whether soil properties reach a steady state. In this study, a timespan of 50 years was selected, to show details in the initial stages of development and in the agricultural phase. With longer timespans, for example 500 years, the disturbances and recovery of soil properties might be averaged out and a steady state might become apparent for quickly developing soil

properties, such as SOM stocks. Slowly developing properties, such as depth to Bt horizons, might require even longer timespans before the disturbance and recoveries are averaged out. The selection of the timespan for the evaluation thus determines to a large extent the level of detail and behavior of the evolutionary pathways. In order to derive meaningful results from the evolutionary pathways, the timespans should be chosen based on the research question, the time period of interest, the expected rate of soil property change and the rate of disturbance cycles.

The findings suggest that soils do not necessarily evolve to mature climax soils, but rather that soil patterns diverge to stable complexity. The spatial configuration of soil properties might change, but the pattern characteristics stay the same. This complexity is a function of disturbance cycles in the landscape, such as tree throw occurrences or tillage practices (Phillips, 2017). When disturbance rates are high, as is the case in intensively managed landscapes, the complexity of patterns of slowly responding soil properties keeps changing, indicating a transient state, while patterns of faster responding soil properties might

reach a new steady state of stable complexity. A simple distinction between natural progressive and agricultural regressive soil development (e.g., Sommer et al., 2008) does not do justice to these complex responses of different soil properties to natural or anthropogenic forcing. The response of soils to local disturbances or sudden shifts in boundary conditions depends on the rates of erosion and the ability of soil properties to recover from these disturbances. Therefore, assessment of evolution of soils and landscapes should be done per individual process or soil property rather than for the entire soil and should be performed

on a landscape scale rather than a pedon scale.

Understanding how different soil properties, both manageable and inherent, respond to different natural and anthropogenic forcing is essential for understanding how ecosystem functions of soils are affected by certain management (Dominati et al., 2010). SLEMs and evolutionary pathways can help the scientific community to develop and adjust sustainable land management practices, when the models are validated and there is confidence on the functioning of the model. This can for

example support the evaluation of spatial and temporal effects of soil management practices for promoting carbon uptake in the soil or reducing land degradation (Minasny et al., 2017; Smetanová et al., 2019).

## 4.2 Applicability for (soil-)landscape evolution modelling

The continuous development of numerical models and quantitative data analysis techniques help increase our understanding of complex soil development under changing boundary conditions (Krasilnikov and Targulian, 2019), the functions that soils provide (Vogel et al., 2018) and improve communication of complex soil science with policy makers (Turner, 2021). Evolutionary pathways are a novel and promising way to analyze and visualize large data output into understandable trends. This work shows the potential of evolutionary pathways for summarizing large data output from numerical soil-landscape evolution models into manageable and understandable trends. This evaluation on a landscape scale considers the overall development of the soil landscape with the SLDS, as well as the variation in the soil landscape with the complexity. By combining these two properties, a combined spatiotemporal overview of the model results is provided, while, traditionally, model output is presented only spatially (e.g., transects) or only temporally (e.g., development of a certain soil profile, time series).

SLEMs are generally reduced-complexity models. This means that the process formulations and drivers are simplified, to facilitate parametrization and simulation of multiple processes, while reducing data requirements and calculation time (Hunter et al., 2007; Temme et al., 2011; Kirkby, 2018). Also, some processes can be left out, such as weathering processes in this study. The current simplified model and its parametrization sufficed to simulate the selected properties for the analysis: SOM stocks and depth to Bt horizons. The inclusion of weathering processes in the simulations might have resulted in more detailed soil properties, but its effect on the spatial and temporal patterns, and thus the evolutionary pathways, would likely be minor. When evaluating other soil properties, a more complex or extensive model might be required. Examples of other properties that can be described with evolutionary pathways are other inherent or manageable soil properties that the model puts out, but also simple topographic properties such as slope profiles, more complex topographic properties such as topographic position index, topographic wetness index or flow accumulation, or output maps of elevation change due to different soil erosion processes. With that, the method is applicable to the results from other (soil-)landscape evolution models or erosion models as well (Temme et al., 2017). Calculating evolutionary pathways is not even limited to soil or geomorphological systems (Phillips, 2016). As long as quantitative spatial and temporal data of model output is present, evolutionary pathways can be determined. The easily calculatable complexity and SLDS already provide new valuable insights into changes of the state and heterogeneity of soil and landscape patterns. The calculations of evolutionary pathways are of course not limited to the mean and standard deviation of these patterns. Other, more complex, statistics can provide additional insights in other properties of soil patterns. For example, changes in the parameters of semivariograms, i.e. the nugget, sill and range, can provide insights in the development of local and regional spatial autocorrelation of soil properties. Changes in correlations between soil properties and terrain properties or boundary conditions, can help shed new light on the spatial and temporal dependence of soil properties on external parameters. These findings might improve statistical soil prediction models by introducing more mechanistic process understanding (Angelini et al., 2016; Ma et al., 2019).

Validation of soil-landscape evolution models is a difficult task, because validation data should cover the parameter domain, spatial domain and temporal domain of these models (Minasny et al., 2015). Field data for some, or all, of these domains are often not available, because data over long timescales is missing, simulated scenarios cover periods of time that are currently not visible or detectable anymore, or SLEMs cover too many different processes and parameters for the available data to calibrate and validate. Here, the evolutionary pathways might provide a solution as well. Temme et al. (2017) suggest that simulated pathways might be constrained using field observations. The complexity or SLDS at a measurable moment in space and time might be used to evaluate the model results for that same moment. When performing explorative modelling, where a simple model is run multiple times with varying inputs (e.g., Larsen et al., 2014), the model results can be converted into evolutionary pathways that can be rejected or accepted using field evidence. These are two ways to ensure a better connection between model and field data, which is currently lacking for most soil-landscape evolution models (Minasny et al., 2015).

## 5    Conclusions

In this study, I used evolutionary pathways to analyze and visualize spatial and temporal trends in previously published results from a soil-landscape evolution model. The evolutionary pathways indicate progressive or regressive soil development and convergence or divergence of the soil pattern, as a consequence of (non-linear) changes in the drivers of soil and landscape development. The evolutionary pathways can be linked to real-world examples of soil development and soil complexity.

Evolutionary pathways are not limited to the examples presented in this study, but can be applied to a wide range of soil pattern statistics and soil and terrain properties. With this, evolutionary pathways provide a promising tool to visualize soil model output, not only for studying past changes in soils, but also for evaluating future spatial and temporal effects of soil management practices in the context of sustainability.

## 6    Competing interests

The author declares that he has no conflict of interest.

## 7    Code availability

See Van der Meij et al., (2020) for model code availability.

## 8    Acknowledgements

This paper forms the completion of my PhD on soil and landscape evolution in the Anthropocene. I want to thank my supervisors Prof. Dr. Arnaud Temme, Prof. Dr. Michael Sommer and Prof. Dr. Jakob Wallinga for their supervision and support during my PhD.

I thank the reviewers and editor for their valuable feedback on the manuscript.

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
