# Peer review of "Evolutionary pathways in soil-landscape evolution models"

_SOIL, 2021_

## Author Response (AR1)

**Author's response**

Dear editor,

Thank you for handling my submission to SOIL. I have now finished the revised manuscript, where I addressed all of the reviewer's comments. I attached a copy of my response to each individual comment below. Here I will list the most important changes to the manuscript, based on the major comments of the reviewers. The line numbers refer to the revised manuscript.

- Based on comment 9 of Reviewer 2 (see below), I reduced the timespan over which the evolutionary pathways were calculated from 500 years to 50 years, to provide more detail in the initial development of soils in the natural phase (line 123-124). By reducing the timespan, the results in Figures 3 and 4 show more detail in the evolutionary pathways and required some adjustments to the interpretations. I rewrote the Results and parts of the Discussion to describe the new Figures, while keeping in mind the comments of Reviewer 2 on the original Sections. Next to that, I added a discussion on choosing the timespan for calculating evolutionary pathways (lines 210-218). Based on the reviewer comments, I also clarified or rewrote other parts of the Discussion.
- I extended the model description, by providing basic information of the model architecture (lines 59-64), giving a more detailed description of the simulated processes (lines 75-91) and adding a Figure showing the model architecture and inputs (Figure 1, lines 70-74). This was necessary to better understand the model results and evolutionary pathways, without reading the paper where the model is presented (Van der Meij et al., 2020).
- I introduced the concept of inherent and manageable soil properties, to illustrate how soil-landscape evolution models (SLEMs) can contribute to designing sustainable soil management practices and to better distinguish between properties that are simulated by the model (lines 30-32, 100-102, 230-235).
- I reduced the emphasis on evolutionary pathways being a suitable communication tool towards policy makers, because reviewer 1 remarked that the evolutionary pathways are still difficult to understand. Next to that, I specified for which community evolutionary pathways might be useful (scientific community, line 232).

With these changes, I think the manuscript has greatly improved and I hope it is now suitable for publication.

With best regards,

Marijn van der Meij

**Reply to reviewer 1**

Dear reviewer,

Thank you for your extensive review of my manuscript and suggested points for improvement. I especially appreciate the concept of inherent and manageable soil properties, that you address in your first comment. This distinction helps me to indicate how soil-landscape models (SLEMs) can support current-day soil management. This is not only valuable for this paper, but also for other works that I have planned.

I respond to your comments below. Your comments are printed in italic and with bullet points.
* * *
- *This short manuscript compares mechanistic soil landscape evolution modelling (SLEM) to the theoretical evolutionary pathway approach and discusses the complementarity of the two approaches with as initial idea that evolutionary pathways could be an efficient tool to summary the modelling results and thus better communicate. The demonstration is made on modelling results from an already published study (van der Meij et al., 2020) that are reinterpreted here in term of evolutionary pathways for two soil characteristics: the depth of the Bt-horizon and the soil organic matter (SOM) stock. The work demonstrates that evolutionary pathways of these two soil characteristics differ due to differences in dynamics of these two characteristics. This idea could however have been pushed forward clearly introducing another concept of the soil sciences: the inherent and manageable soil characteristics as defined by Dominati et al. (2010) among others; depth of the Bt-horizon representing the inherent characteristics and SOM the manageable ones.*

**Response:** Distinguishing between inherent and manageable soil properties is indeed valuable when comparing different soil properties, especially in the context of ecosystem services. SLEMs are able to simulate the development of inherent properties, such as soil texture, soil depth and slope profiles, as well as some manageable soil properties, such as carbon stocks. I will add introduce this distinction in a few sentences in the Introduction and elaborate on the use of evolutionary pathways for evaluating inherent and manageable properties in the Discussion.

- *In addition, the potentiality of evolutionary pathways to be used as a communication tool should be specified, notably by clarifying toward which community this communication is thought to. Indeed evolutionary pathways are a convincing tool to sum up the SLEM results but they are not easy to understand and thus probably not suitable to communicate with most of the soil end-users.*

**Response:** As you rightly point out, I indeed didn't specify to which audience the SLEM output should be communicated, while I do mention in the abstract that evolutionary pathways are promising tools to communicate soil model output.

The work in this paper, where I use evolutionary pathways for analysing soil and landscape evolution, is probably most interesting for the scientific community that wants to get a better understanding of soil development and soil-landscape variability. Evolutionary pathways can also be used to analyse the results of other soil models, for example models that evaluate the effect of different soil and land management strategies on manageable soil properties. In that case, the results will also be of interest to policy makers and designers of soil and land management strategies.

I will briefly elaborate on the target audience in the Discussion of the manuscript and link this Discussion to the point raised above, about inherent and manageable soil properties.

*Out of these two main comments, the manuscript is very well written and easy to read and follow. I added detailed comments below to be addressed. Once these comments addressed, I think that this manuscript is worth publishing in SOIL.*

**Detailed comments**

*Method section*

- *Model study. This is the summary of a study published elsewhere. The presented model does not contain weathering processes at all, which can be a severe limitation when used over 15 000 years. I recognised that the aim of this manuscript is not to discuss the model. I nevertheless think that this limitation should be mentioned in this section and discussed in the evolutionary pathway section of the discussion line 153.*

**Response:** In the modelling study, we decided to adopt a simplified advection-diffusion equation for the clay translocation process, that matched the reduced complexity of the model. This process description was able to reproduce measured clay-depth profiles and did not require the simulation of weathering processes. For future modelling studies, we will reconsider including weathering processes, because weathering can indeed be an important process for generating clay particles through breakdown of particles or clay neoformation, especially over such long timescales.

I will elaborate in Section 2.1 on the processes that were and were not included in the model, also following comments from Referee #2, and I will mention the possible effects of leaving out weathering in the Discussion.

- *Evolutionary pathway. Please add the units used in the different equations*

Response**:** The units of the equations are the change in the property units over time. For the SOM stocks, this means the units are kg m$^{-2}$ a$^{-1}$ and for the depth to Bt the units are m a$^{-1}$. I will add a sentence at the end of Section "2.2 evolutionary pathways" where I mention the units.

*Results*

- *Page 5 line 119, replace "the complexity" by "it" to avoid useless repetitions.*

**Response:** I will change the wording following your suggestion.

- *Page 7 lines 133-137, these statements seem to be true mainly for the depth of the Bt-horizon.*
- *Same page, lines 137-138, on the opposite, this statement seems to be true mainly for SOM.*

**Response:** Based on your comments and the comments of Referee #2, I will rewrite this Section. I will better explain the differences between the natural and agricultural phase for both soil properties.

- *Same page, sentence lines 142 to 145, please refer to figure 2 (more appropriate than figure 3).*

**Response:** I will add a reference to Figure 2 for the first sentence of this comment. For the second sentence, I believe Figure 3 better illustrates the point that I want to make. Moreover, I will move this sentence to the Discussion, based on comments of Referee #2.

- *Same page, lines 145-146, add "for SOM" after "new steady state" and "for the depth to Bt" after "a steady rate of change".*

**Response:** I will add the text you suggested.

*Discussion*

- *Page 8 line 161. A reference to results found in van der Meij et al. (2020) is made. This should be extended to be clearer to the reader as I had to go to that paper to understand. It presents semi-variogramm and when the statement seemed clear to be for SOM, it is not so much the case for the depth of the Bt horizon.*

**Response:** I will remove the reference to the Figure in the accompanying paper and rephrase the sentences, so that they are now based on the results from this study.

- *Same page lines 172-173, this statement mainly apply to manageable soil characteristics according to the present study. See comment above.*

**Response:** I will address the effect of humans on manageable and inherent soil properties after this sentence.

*Figures*

- *Fig. 1. Please don't use the acronym in the figure title so it can be understood by itself*

**Response:** I will write the acronym SLDS in full in the Figure caption.

***Reference cited***

*Dominati E, Patterson M, Mackay A (2010) A framework for classifying and quantifying the natural capital and ecosystem services of soils. Ecological Economics 69, 1858-1868. doi:10.1016/j.ecolecon.2010.05.002*

**Reply to reviewer 2**

Dear reviewer,

Thank you for the review of the paper and pointing out some inconsistencies and unclarities in the analysis of the data.

I addressed your comments below one by one. Your comments are printed in italic. I also numbered them, which made it easier to refer back to a response on an earlier comment. I hope that the suggested edits will make the paper better readable and understandable.
* * *
*This short forum paper applies the concept of evolutionary pathways to the results obtained with the use of soil-landscape evolution models (SLEMs). In that specific paper, the author used outputs from the HydroLorica SLEM. The paper is short, well written. The authors are convincingly presenting this application of the evolutionary pathways concept, with a very interesting second part of the discussion section to broaden the scope of this paper. It uses extensively the results obtained in an other previous paper (Van der Meij et al., 2020) to calculate two types of evolutionary pathway parameters at a given timestep, that are accounting for time and space variations : the soil-landscape development stage (SLDS) using depth to the Bt horizon and SOC stocks as proxies, and the complexity using the standard deviations of these paramaters as proxies. These two indicators are simple to calculate, and allow to convincingly summarize the soil/landscape evolution. Moreover, SLDS and complexity can be easily applied to other model results, which shows their versatility and generalization potential.*

*Overall, I think this forum paper is worth publishing, providing a few issues are dealt with:*

1. *The main issue deals with the interpretation of figures 2 and 3 that does not always seem to me to be correct, and require many changes in the text (lines 104-145 and some parts of the discussion, see below). This should however not be a difficult task.*

**Response:** Thank you for pointing out the errors and inconsistencies in the interpretation of the results. I will address all comments you make below in the review and restructure parts of the discussion to improve readability and remove errors or inconsistencies in the interpretation of model results.

2. *The second issue deals with the difficulty to read this paper without reading extensively the previous paper in which the HydroLorica model is applied. As it is, this paper does not always give sufficient details about the structure of the HydroLorica model, to be able to understand the explanations given for the evolution of both SLDS and complexity parameters in the discussion section. I would therefore recommend to give a bit more details about the structure of the model, and specifically some details about the components of the model that indeed have an influence on the behaviour of the SLDS and complexity evolutionary pathway parameters.*

**Response:** This paper is indeed missing some information that is needed to understand the model behaviour and the results. I hoped that listing the original modelling study as companion paper, the information might be easy to retrieve for the readers. But, as you correctly point out, some results cannot be interpreted correctly using the information in this paper. Therefore, I will describe the model study in more detail in Section 2.1 and add a Figure that shows the model

structure and required input data. I hope that these additions will make the paper better understandable to the readers, without having to read the original paper.

*Detailed comments :*

3. *Line 68 : can the author present a figure with the topography of the artificial catchment (contour lines) ? This would allow to have a better understanding of the initial conditions.*

**Response:** I will add a Figure showing the input data and the model structure (see response at comment 2) to clarify how the model works and what it requires in terms of input.

4. *line 77 : the depth to Bt horizon is used as a SLDS parameter. Does the author infer that the conditions for clay translocation processes are met over all the catchment area ? It if iyt the case, it would be worth to write it. I would actually need more explanation about the processes that provoke clay translocation in the model, to actually understand why this translocation occurs in any topographic situation.*

**Response:** The model indeed simulates clay translocation in every location. The only limitations for clay translocation is the availability of transportable clay and infiltrating water. The model does not include other factors or processes that could limit clay translocation, such as stagnating water levels or net upward flow of water. I will mention that clay translocation is possible everywhere in the model, together with a more extensive description of the clay translocation process.

5. *lines 106-107 : replace 'than the dry sceanario' by 'for the wet scenario'.*

**Response:** I will rephrase the sentence to your suggestion.

6. *lines 107-110 : this part is not clear to me and would deserve to be rewritten, as some facts are repeated (=> slight increase of SLDS for the dry sceanrio). Moreover, according to figure 2A, I do ot completely agree with what is written : I see the complexity for the wet scenario increasing again very shortly after the onset of agriculture, and not 250 years into the agricultural phase (some 50 to 100 years after according to Figure 3C ?). The evolution of SLDS for the dry scenario does seem to have a more complex behaviour than just a slight increase after the onset of agriculture : according to figure 3A, there is first a significant decrease of the depth to Bt.*

**Response:** I will rephrase this Section, to give more detailed explanation of the changes in evolutionary pathways, while also correcting the errors you found.

7. *lines 115-119 : again, this paragraph is not clear to me and needs rewriting. Lines 116-117 : the part of the sentence 'due to higher SOM input in grassland systems and less redistribution processes' should be transfered to the discussion section. Moreover, I do not think data presented in this paper justify this statement.*

**Response:** I will remove sentence 116-117 from the Results Section. This point is also already addressed in the Discussion. With the extra information that I will provide on model structure and inputs (see response comment 2), where I will also mention the increased erosion rates and changes in SOM input, I can justify making this statement and explain why the evolutionary

pathways change as they do. Next to that, I will also rewrite the rest of the Section, to clarify what is visible in Fig. 2B.

> 8. *line 125, figure 3 caption : please specify that each point in the plot corresponds to a 500 years timestep for the natural phase, and 50 years timestep for the agricultural phase.*

**Response:** I will specify the timesteps over which the evolutionary pathways are calculated in the caption of Figure 3.

> 9. *lines 132-134 : looking at figure 3D, despite what is stated, it is not at all obvious for SOM that natural evolutionary pathways in the wet scenario are more divergent and converge more slowly to the origin of the graph. I however agree this is the case for depth to Bt horizon. This should be corrected. I guess we can not see any difference for SOM for the natural evolution between both scenarios because steady state is reached faster than the 500 years timestep of the model. Perhaps we would have more information for SOM stocks evolution in the natural phase considering smaller timesteps at incipient stages of soil development, due to the inherent quick response of SOM dynamics compared with depth to Bt horizon.*

**Response:** Based on your comments, and the comments of Referee #1, I rewrote this Section. Some of the points that I address are indeed not clearly visible in the Figure. I will try to visualize this better, for example by adding a detail of the origin of the graphs that shows the slow convergence towards the origin of the graph, or by increasing the temporal resolution of the data points for the first 500 years of natural soil development.

> 10. *lines 138-139 : looking at figure 3, while the magnitude of change for SOM is clearly higher for the first step after cultivation, it is not the case for depth to Bt horizon. Please consider modifying.*

**Response:** I will rephrase this and mention that, for depth to Bt horizon, the changes in the agricultural phase are similar to those in the start of the natural phase.

> 11. *lines 140-141 : again, this sentence should be transferred to the discussion section. I indeed do not understand why "potential clay translocation in clluvial positions" would induce shallower depths to Bt horizons. Indeed I would infer this would be the opposite.*

**Response:** As indicated in the response to comment 7, this point will be addressed in the Discussion. In the case of clay translocation in colluvial deposits, there can be an increase in clay content in the upper layers of te colluvial soil, that can be classified as a Bt horizon. In that case, this layer will be recognized as the Bt horizon, leading to shallower bt horizons and consequently regressive pedogenesis. I will address this point in the Discussion as well.

> 12. *lines 142-145 : consider transferring these sentences in the discussion section ?*
> 13. *lines 145-146 : change to "all pedgenic pathways either kept or turned divergent". It is indeed not so obvious for the wet scanario, where the divergent behaviour seems to have taken a bit more time, especially for SOM.*

**Response:** I will move these sentences to the discussion and specify when the evolutionary pathways of the two soil properties turn divergent.

> 14. *- lines 151-152 : I do not see what to conclude from this statement, where increased water erosion and clay translocation have opposite effects on the depth to the Bt horizon.*

**Response:** This statement was about divergent soil development. Water erosion and increased clay translocation indeed have opposite effects, causing an increase in the complexity of the soil pattern. I will specify this in the text.

15. *lines 152-155 : I do not understand. The author states that a steady state is not reached in the natural setting for the wet scanario, however, figure 3 shows the opposite : at the end of the natural phase, a steady state is reached, illustrated by the position of the point at the origin of the graph. I understand there is no steady state that is reached for a given point in the landscape, but a strady state is reached on average.*

**Response:** In the wet scenario, the soil properties take longer to approach a steady state compared to the dry scenario. For the depth to Bt in the wet scenario, the steady state is not reached. Instead, the points remain scattered around the origin, which indicates there are still changes in the properties that prevent the reaching of a steady state. This is caused by the tree throw process in the model, that locally reworks the soil. The process of clay translocation is apparently not fast enough to restore the soil pattern over the time span of analysis (500 years), causing fluctuations around the origin of the graph. This is not the case for SOM stocks, which apparently recover quicker than the depth to Bt. I will specify this in the text and illustrate it with a detailed view of the origin of the plots, that shows the scatter around the origin (see also the response to comment 9).

16. *lines 156-158 : the figure 3 does not show in an obvious manner that in the wet scanario, the equilibrium takes longer to be reached than for thre dry scenario (and not at all for SOM stocks).*

**Response:** These patterns are indeed not well visible for SOM stocks. I will specify that this comment is about depth to Bt horizon and change the reference to Figure 2, where these patterns are better visible.

17. *lines 161-163 : I do not think reference can be made to a figure in an other paper to explain processes that are described in this paper. It is necessary to have more information in this paper*

**Response:** I will remove the reference to the other paper and rephrase the statement using information that is presented in this paper.

18. *lines 163-164 : I do not understand this statement. Can the author justify it ?*

**Response:** I will remove this unclear sentence and rephrase the sentence before to clarify this statement.

19. *lines 169-171 : I do not understand this statement. It has been shown on Fig. 3 that a steady state is reached in the natural phase for both SOM stocks and depth to Bt. Why is it stated here that this steady state is only reached for SOM stocks ?*

**Response:** See my response to comment 15 for more details on this point. I will restructure this part of the Discussion, so that all discussions about steady states are together and it is easier to explain how the soil properties behave in the model.

20. *lines 171-172 : it is not the case for delta SLDS for the depth to the Bt horizon, if the first stage of soil development is accounted for (figure 3).*

**Response:** I will rephrase this sentence following the outcomes of the proposed adjustments to Figure 3 as proposed in the response to comment 9.

**References**

Van der Meij, W. M., Temme, A. J., Wallinga, J., and Sommer, M.: Modeling soil and landscape evolution–the effect of rainfall and land-use change on soil and landscape patterns, 6, 337–358, https://doi.org/10.5194/soil-6-337-2020, 2020.

---

## Author Response (AR2)

**Author's response**

Dear editor,

I want to thank you and the reviewers for the valuable feedback on the manuscript. It has become a very nice paper.

In this last version I processed last textual remarks made by Sebastién Salvador-Blanes. I adopted all remarks, except for the comment on the colours in Fig. 3. I think that this suggestion will make the legend of the Figure more difficult to understand. Instead, I added an extra line in the captions, clarifying the meaning of the colours.

With best regards,

Marijn van der Meij